# The pro-regenerative effects of hyperIL6 in drug-induced liver injury are unexpectedly due to competitive inhibition of IL11 signaling

Jinrui Dong[1], Sivakumar Viswanathan[1], Eleonora Adami[1], Sebastian Schafer[1], Fathima F Kuthubudeen[1], Anissa A Widjaja[1], Stuart A Cook[1,2,3]*

[1]Cardiovascular and Metabolic Disorders Program, Duke-National University ofSingapore Medical School, Singapore, Singapore; [2]National Heart Research Institute Singapore, National Heart Centre Singapore, Singapore, Singapore; [3]MRC-London Institute of Medical Sciences, Hammersmith Hospital Campus, London, United Kingdom

**Abstract** It is generally accepted that IL6-mediated STAT3 signaling in hepatocytes, mediated via glycoprotein 130 (gp130; IL6ST), is beneficial and that the synthetic IL6:IL6ST fusion protein (HyperIL6) promotes liver regeneration. Recently, autocrine IL11 activity that also acts via IL6ST but uses ERK rather than STAT3 to signal, was found to be hepatotoxic. Here we examined whether the beneficial effects of HyperIL6 could reflect unappreciated competitive inhibition of IL11-dependent IL6ST signaling. In human and mouse hepatocytes, HyperIL6 reduced N-acetyl-p-aminophenol (APAP)-induced cell death independent of STAT3 activation and instead, dose-dependently, inhibited IL11-related signaling and toxicities. In mice, expression of HyperIl6 reduced ERK activation and promoted STAT3-independent hepatic regeneration (PCNA, Cyclin D1, Ki67) following administration of either IL11 or APAP. Inhibition of putative intrinsic IL6 trans-signaling had no effect on liver regeneration in mice. Following APAP, mice deleted for *Il11* exhibited spontaneous liver repair but HyperIl6, despite robustly activating STAT3, had no effect on liver regeneration in this strain. These data show that synthetic IL6ST binding proteins such as HyperIL6 can have unexpected, on-target effects and suggest IL11, not IL6, as important for liver regeneration.

*For correspondence:
stuart.cook@duke-nus.edu.sg

## Introduction

The liver has an extraordinary capacity to regenerate in response to injury. Replication of hepatocytes in midlobular zone two underlies liver regeneration (*Wei et al., 2021*), with a large number of cytokines and growth factors implicated as mitogens (*Michalopoulos and Bhushan, 2021*). Interleukin 6 (IL6), a member of the larger IL6 family of cytokines, binds with high affinity to its alpha receptor (IL6R) to signal in cis via glycoprotein 130 (gp130; IL6ST) and STAT3. Of all the cytokines implicated in liver regeneration, *IL6* is believed to be a predominant auxiliary mitogen (*Michalopoulos and Bhushan, 2021*; *Schmidt-Arras and Rose-John, 2016*). This belief is anchored on a seminal study performed in mice globally deleted for *Il6*, which exhibit reduced STAT3 activity and lesser liver regeneration following injury (*Cressman et al., 1996*).

It is thought that IL6 can bind to a soluble form of its receptor (sIL6R) to signal in trans to activate IL6 signaling in cells that express IL6ST but low/or no IL6R (*Schmidt-Arras and Rose-John, 2016*). This led to the design of an artificial fusion protein composed of a truncated form of human IL6R linked to human IL6 (HyperIL6). HyperIL6 stimulates STAT3 signaling up to 1000-fold stronger than the

respective separate molecules with high affinity for IL6ST (*Fischer et al., 1997*; *Peters et al., 1998*). The HyperIL6 superagonist can reverse fulminant liver failure due to toxin-induced liver damage (*Galun et al., 2000*; *Hecht et al., 2001*) and stimulate liver regeneration after partial hepatectomy (*Peters et al., 2000*). The pro-regenerative activity of HyperIL6 has also been observed in the spinal cord (*Leibinger et al., 2021*), optic nerve (*Fischer, 2017*), kidney (*Nechemia-Arbely et al., 2008*), and heart (*Matsushita et al., 2005*).

We recently found that IL11, a little studied IL6 family protein, is hepatotoxic and important for NASH pathologies (*Dong et al., 2021*; *Widjaja et al., 2019*). Furthermore, in a recent study of N-acetyl-p-aminophenol (APAP)-induced liver injury, IL11 was shown to activate NOX4, ERK and JNK and impede liver regeneration (*Widjaja et al., 2021*). Interestingly, this study demonstrated that synthetic, IL6ST-binding proteins can compete with endogenous IL11 for binding to IL6ST and reduce APAP-induced hepatotoxicity. In light of this new data, it is possible that HyperIL6 could compete with IL11:IL11RA complexes for binding to IL6ST and thus inhibit maladaptive IL11 signaling. Here we investigated whether the mechanism of action of HyperIL6 in liver regeneration is due to inhibition of IL11 signaling and, in contrast to the accepted paradigm, independent of STAT3 activation.

## Results

### STAT-independent HyperIL6 activity inhibits APAP- and IL11-induced hepatocyte cell death

To test our hypothesis, we studied APAP-induced hepatotoxicity. APAP poisoning is a common cause of liver damage, associated with impaired liver regeneration (*Bernal and Wendon, 2013*). In primary human hepatocytes cultures, incubation with APAP for 24 hr caused cell death in approximately 40 % of cells (*Figure 1A,B*, *Figure 1—figure supplement 1A, B*). Inhibition of IL11 signaling using a neutralizing IL11RA antibody (X209) reduced ERK, JNK, and NOX4 activity and cell death (*Figure 1A–C*). These phenotypes were mirrored by antibody-based neutralization of IL6ST. HyperIL6 also inhibited APAP-induced cell death, and this was associated with increased STAT3 phosphorylation and lesser ERK, JNK, and NOX4 activity (*Figure 1A–C*; *Figure 1—figure supplement 1A,B*).

In human hepatocytes, HyperIL6 markedly induced STAT3 phosphorylation but had minimal effect on ERK and no effect on AKT (*Figure 1D*). Inhibition of IL11 signaling with X209 or anti-IL6ST reduced APAP-induced reactive oxygen species (ROS) and maintained cellular glutathione (GSH) levels, which was also true for HyperIL6 (*Figure 1E,F*, *Figure 1—figure supplement 1C*). These initial studies show that HyperIL6 uniquely activates STAT3 but inhibits APAP-induced signaling and cellular phenotypes similarly to neutralizing IL11RA or IL6ST antibodies (*Figure 1A–C and E–F*, *Figure 1—figure supplement 1C*).

We then examined the functional relevance of HyperIL6-induced STAT3 activation in hepatocytes exposed to APAP. Interestingly, S3I-201 (a STAT3 inhibitor; iSTAT3) had no effect on the protection afforded by HyperIL6 despite inhibiting STAT3 activation (*Figure 1A, F and G*). Furthermore, S3I-201 had no effect on HyperIL6-induced cell death, ROS induction, or GSH depletion. At the signaling level, S3I-201 inhibited STAT3 activation, but not ERK or JNK phosphorylation nor NOX4 upregulation (*Figure 1A–C and E–G*, *Figure 1—figure supplement 1C,D*). These experiments suggest that the beneficial effects of HyperIL6 are unrelated to STAT3 activity but instead reflect competitive inhibition of IL11 signaling (*Figure 1H*).

We then examined whether HyperIL6 could directly inhibit IL11 signaling in hepatocytes. Incubation of hepatocytes with IL11 resulted in ERK, JNK, and NOX4 activation and cell death, as expected and similar to that seen with APAP (*Figure 1I,J*, *Figure 1—figure supplement 1E,F*; *Widjaja et al., 2021*). HyperIL6 dose-dependently inhibited IL11 signaling and toxicity that was independent of STAT3 phosphorylation and could be titrated away by the addition of soluble IL6ST (sIL6ST) (*Figure 1I,J*, *Figure 1—figure supplement 1E,F*). We went on to show that the protective effects of HyperIL6 on APAP toxicity in human hepatocytes could be dose-dependently inhibited by the addition of sIL6ST. We confirmed again that the protective effects of HyperIL6 were STAT3 independent and instead related to inhibition of IL11 signaling (*Figure 1—figure supplement 1G,H*).

In binding assays, HyperIL6 bound to IL6ST with a similar dissociation constant as an IL11:IL11RA construct (HyperIL11) ($K_D$ = 1 nM and 0.95 nM, respectively), whereas IL6 alone did not bind to IL6ST (*Figure 1—figure supplement 2A–C*). These data would be consistent with competitive inhibition of

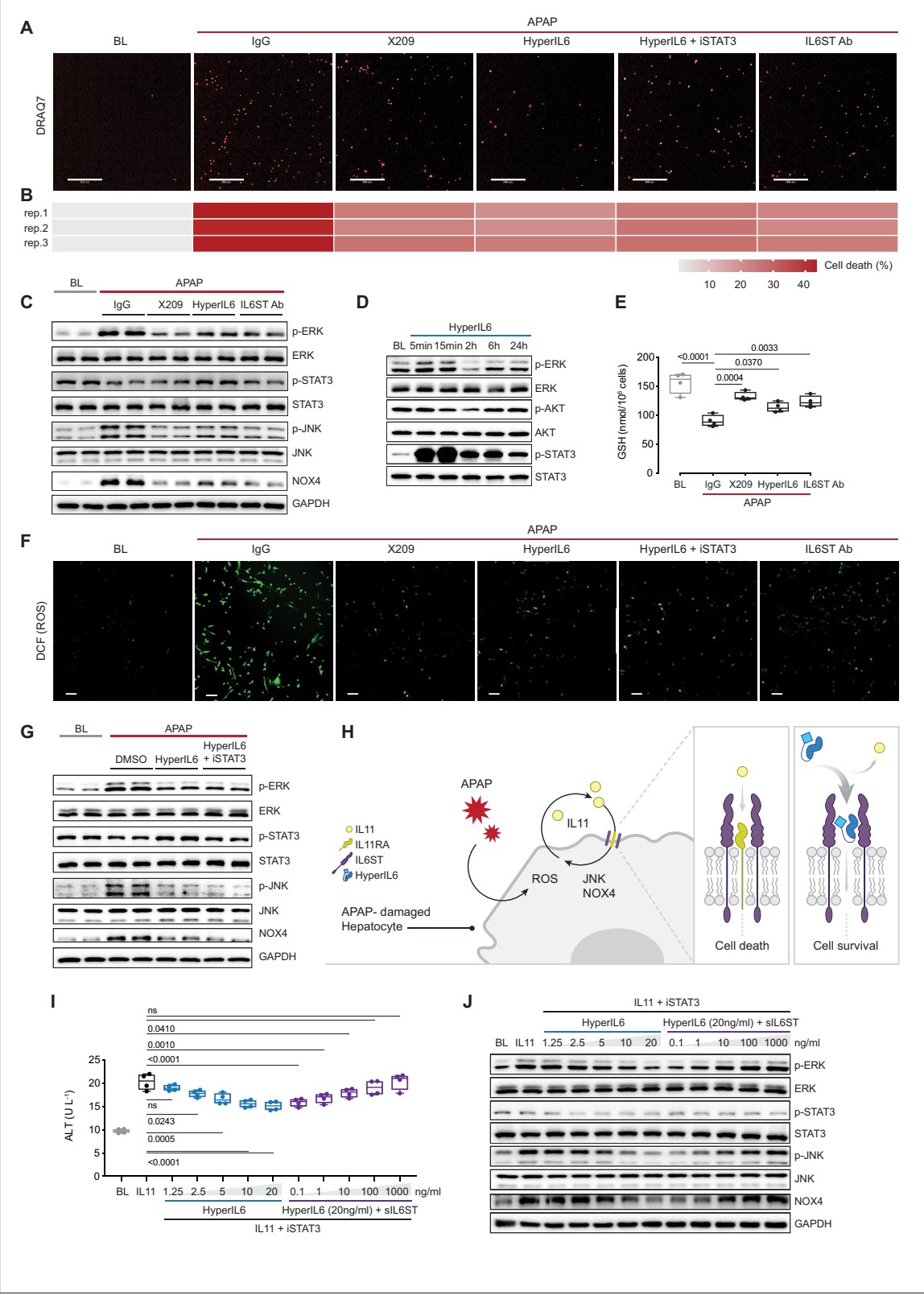

**Figure 1.** STAT-independent HyperIL6 activity inhibits APAP- or IL11-stimulated cell death through competitive binding to the IL6ST co-receptor. (**A**) Representative fluorescent images and (**B**) quantification of DRAQ7 staining for cell death (scale bars, 200 μm) (n = 3 independent experiments, 23 images per experiment) in APAP (20 mM) treated hepatocytes in the presence of IgG (2 μg/ml), DMSO, anti-IL11RA (X209, 2 μg/ml), HyperIL6 (20 ng/ml), HyperIL6 supplemented with iSTAT3 (S3I-201, 20 μM), or anti-IL6ST (2 μg/ml). (**C**) Western blots showing phospho-ERK, ERK, phospho-STAT3, STAT3,

*Figure 1 continued on next page*

*Figure 1 continued*

phospho-JNK, JNK, NOX4, and GAPDH levels in APAP-treated hepatocytes in the presence of IgG, X209, HyperIL6, or anti-IL6ST. (**D**) Western blots of phosphorylated ERK, AKT, and STAT3 protein and their respective total expression in hepatocytes in response to HyperIL6 stimulation. (**E**) GSH levels (n = 4) in APAP-treated hepatocytes. (**F**) Representative fluorescent images of DCFDA (2',7'-dichlorofluorescein diacetate) staining for ROS detection (scale bars, 100 μm) (n = 4 independent experiments, 10 images per experiment) in APAP-treated hepatocytes. (**G**) Western blots showing ERK, STAT3, and JNK activation status, NOX4 protein expression in APAP-treated hepatocytes in the presence of DMSO, HyperIL6, or HyperIL6 supplemented with iSTAT3. (**H**) Proposed mechanism for competition of IL11 *cis*-signaling and IL6 *trans*-signaling by binding to IL6ST. (**I**) ALT secretion (n = 4) and (**J**) western blots showing ERK, STAT3, and JNK activation status, NOX4 protein expression by rhIL11 (10 ng/ml) treated hepatocytes following a dose range stimulation of either HyperIL6 or sIL6ST in the presence of iSTAT3. (**A–G, I–J**) Primary human hepatocytes; (**A–C, E–G, I–J**) 24 hr stimulation. (**E, I**) Data are shown as box-and-whisker with median (middle line), 25th–75th percentiles (box), and min–max values (whiskers), one-way ANOVA with Dunnett's correction.

The online version of this article includes the following source data and figure supplement(s) for figure 1:

**Source data 1.** Raw data, western blot quantification, and fluorescence intensity for panels A–G, I–J.

**Source data 2.** Western blot images (original and annotated) for panels C, D, G, J.

**Figure supplement 1.** STAT-independent HyperIL6 activity inhibits APAP- or IL11-stimulated hepatocyte cell death.

**Figure supplement 1—source data 1.** Raw data, western blot quantification, and fluorescence intensity for panels A–H.

**Figure supplement 1—source data 2.** Western blot images (original and annotated) for panels F–G.

**Figure supplement 2.** Surface plasmon resonance analysis of IL6, HyperIL11, or HyperIL6 binding to IL6ST.

endogenous IL11:IL11RA binding to IL6ST by exogenous HyperIL6 and may explain why IL6 alone is ineffective for liver regeneration (*Nechemia-Arbely et al., 2011*).

## Hepatocyte-specific expression of HyperIL6 prevents APAP-induced liver injury

We next studied the effects of HyperIL6 on APAP-induced liver injury in vivo. Earlier studies used HyperIL6 made from human IL6 and IL6R in the mouse experiments (*Galun et al., 2000*; *Hecht et al., 2001*). This could have unappreciated off-target effects, toxicities, and/or immunogenicity issues as human IL6 and IL6R have limited conservation with mouse orthologs (41% and 53.4%, respectively). Therefore, we examined the effects of recombinant mouse HyperIL6 (rm-HyperIL6) versus recombinant human HyperIL6 (rh-HyperIL6) in the mouse model of APAP injury (*Figure 2A*). We found that both constructs equally reduced serum (alanine transaminase) ALT and (aspartate aminotransferase) AST levels and GSH depletion (*Figure 2B–D*), activated STAT3, and inhibited ERK and JNK phosphorylation (*Figure 2E*). Histology showed both constructs also reduced centrilobular necrosis, pathognomonic of APAP liver damage (*Figure 2F*).

We therefore used species-matched mouse HyperIL6 for hepatocyte-specific HyperIl6 expression studies. Mice were injected with adeno-associated virus serotype 8 (AAV8) encoding either *albumin* promoter-driven mouse HyperIl6 (AAV8-*Alb*-HyperIl6) or one of two controls: AAV8-*Alb*-s*Il6st* or AAV8-*Alb*-Null. AAV8-*Alb*-s*Il6st*, which encodes mouse sIL6ST, provides a second viral control group while probing for effects of endogenous IL6 *trans*-signaling. We compared data from the AAV8-treated mice with a group where we inhibited IL11 signaling by X209 (*Figure 2G*).

The day after APAP (24 hr), mice over-expressing HyperIL6 (*Figure 2H*) or receiving an anti-IL11RA antibody (X209) had lower ALT/AST levels as compared to AAV8-*Alb*-Null group (*Figure 2I,J*). While AAV8-*Alb*-s*Il6st* induced high sIL6ST expression, it had no effect on APAP-induced liver injury (*Figure 2I–K*).

Liver regeneration is associated with a signature of increased Ki67, PCNA, and Cyclin D1 expression (*Sekiya and Suzuki, 2011*), which was apparent 24 hr post-APAP in both HyperIL6-expressing mice and X209-treated mice but not in AAV8-*Alb*-Null+ IgG or in s*Il6st*-expressing mice (*Figure 2K*, *Figure 2—figure supplement 1*). HyperIL6 or X209 partially restored liver GSH levels and inhibited ERK and JNK activation, whereas STAT3 was uniquely activated in HyperIL6-expressing mice (*Figure 2L,M*). Histology revealed typical centrilobular necrosis in APAP-treated AAV8-*Alb*-Null or s*Il6st* expressing mice, which was lesser in mice expressing HyperIL6 or following X209 administration (*Figure 2N*).

These data show that both human and mouse HyperIL6 are protective against APAP-induced liver damage in mice and show that inhibition of IL11 signaling, not activation of STAT3, likely underlies

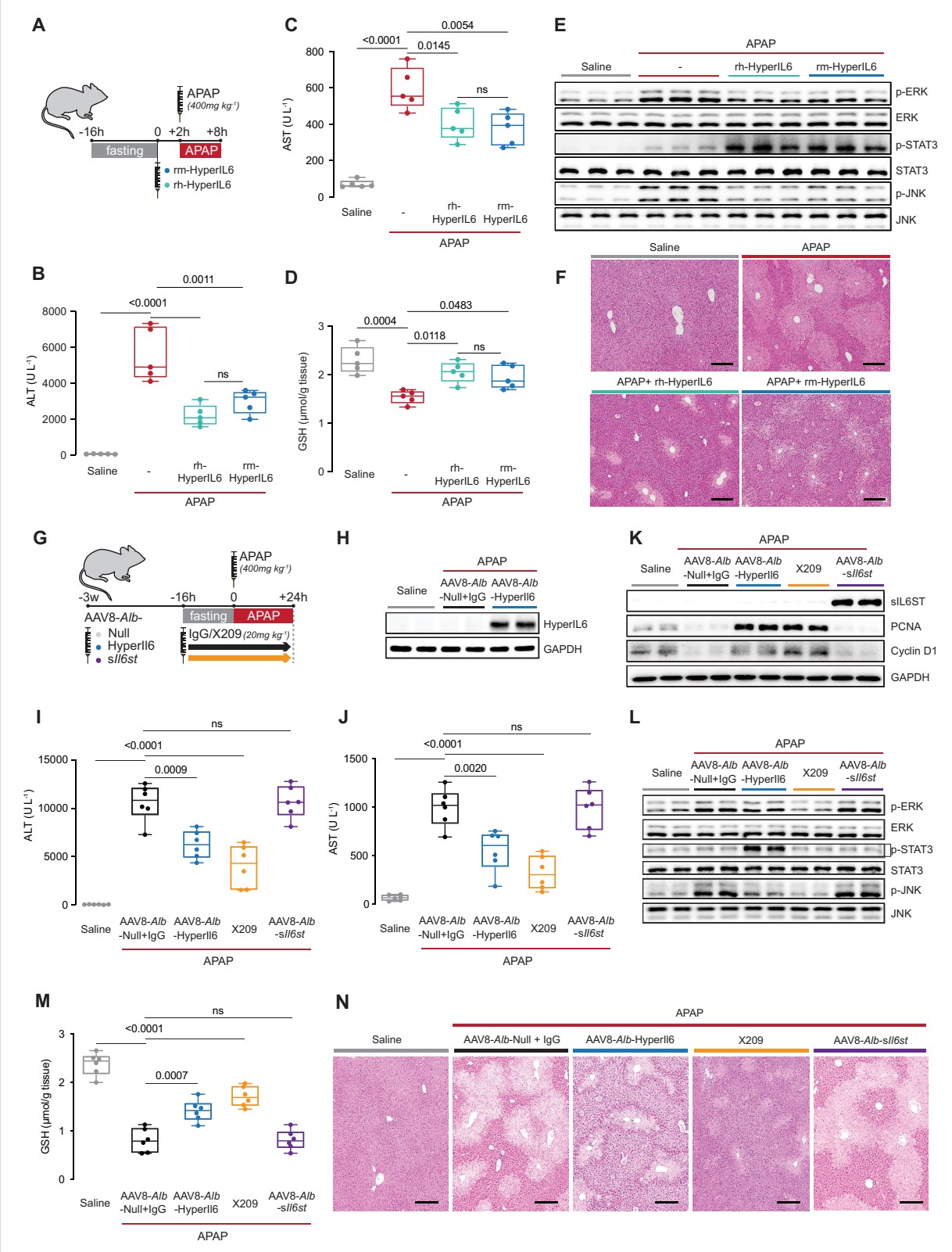

**Figure 2.** Hepatocyte-specific HyperIl6 expression reduces APAP-induced liver injury and phenocopies inhibition of IL11 signaling. (**A**) Schematic of mice receiving rh-HyperIL6 or rm-HyperIL6 (500 μg/kg) administration 2 hr prior to APAP injection; mice were harvested 6 hr post-saline or APAP injection. (**B**) Serum ALT levels, (**C**) serum AST levels, (**D**) hepatic GSH levels, (**E**) western blot analysis of hepatic ERK, STAT3, and JNK activation, and (**F**) representative H&E-stained liver images (scale bars, 50 μm) for experiments shown in (**A**). (**G**) Schematic of APAP-injected mice with hepatocyte-specific

*Figure 2 continued on next page*

*Figure 2 continued*

expression of HyperIl6/s*Il6st* or IgG/X209 administration. Three weeks following AAV8-*Alb*-Null, AAV8-*Alb*-HyperIl6, or AAV8-*Alb*-s*Il6st* virus injection, mice were injected with APAP (400 mg/kg); X209 or IgG (20 mg/kg) was administered at the beginning of fasting period, 16 hr prior to APAP injection; control mice received saline injection; mice were harvested 24 hr post-saline or APAP injection. (**H**) Western blots of hepatic HyperIL6 expression and GAPDH as internal control, (**I**) serum ALT levels, (**J**) serum AST levels, (**K**) western blots showing hepatic levels of sIL6ST, PCNA, Cyclin D1, and GAPDH as internal control, (**L**) western blots showing hepatic levels of phospho-ERK, ERK, phospho-STAT3, STAT3, phospho-JNK, and JNK, (**M**) hepatic GSH levels, and (**N**) representative H&E-stained liver images (scale bars, 50 μm) for experiments shown in (**G**). (**B–D**) N = 5 mice/group; (**I–J, M**) n = 6 mice/group. (**B–D, I–J, M**) Data are shown as box-and-whisker with median (middle line), 25th–75th percentiles (box), and min–max values (whiskers), one-way ANOVA with Tukey's correction.

The online version of this article includes the following figure supplement(s) for figure 2:

**Source data 1.** Raw data, western blot quantification, and necrotic area (%) of H&E-stained liver images for panels B–F, H–N.

**Source data 2.** Western blot images (original and annotated) for panels E, H–K, L.

**Figure supplement 1.** Hepatocyte-specific HyperIl6 expression promotes liver regeneration following APAP-induced liver injury and phenocopies inhibition of IL11 signaling.

**Figure supplement 1—source data 1.** Positive cell counts of Ki67-stained liver images.

HyperIL6 effects. The data also rule out a pro-regenerative effect of putative endogenous *trans*-IL6 signaling.

## The protective effects of HyperIL6 on APAP liver injury are STAT3 independent

To exclude a protective role for STAT3 activation downstream of HyperIL6, we first studied the effects of S3I-201 (10 mg/kg) on HyperIL6-mediated hepatoprotection (*Figure 3A*). Following APAP (6 hr), mice with hepatocyte-specific HyperIL6 expression, either with or without coadministration of iSTAT3, had reduced serum ALT/AST levels, improved hepatic GSH levels, lesser ERK/JNK activity, and diminished centrilobular necrosis (*Figure 3B–E*, *Figure 3—figure supplement 1A*). We observed elevated STAT3 phosphorylation in APAP-treated control mice that was further increased in AAV8-*Alb*-HyperIl6 mice but absent in mice receiving S3I-201 (*Figure 3D*). Thus, the beneficial effects of HyperIL6 on hepatoprotection are STAT3 independent at this early time point of assessment.

Markers of liver regeneration peak some 48 hr following liver injury (*Michalopoulos and Bhushan, 2021*). In addition, APAP liver toxicities can be affected by the administration of dimethyl sulfoxide (DMSO), which we used for S3I-201 stock solutions (*Park et al., 1988*). Therefore, we performed a separate set of experiments to assess regenerative liver phenotypes at 48 hr following APAP and included additional controls to rule out potential confounding effects of DMSO (*Figure 3F*).

Two days (48 hr) after APAP dosing, APAP and APAP+ DMSO treatment groups were indistinguishable with equally elevated ALT/AST, reduced GSH, activated ERK/JNK, diminished PCNA/Cyclin D1/Ki67, and similar patterns of centrilobular necrosis (*Figure 3G–K*, *Figure 3—figure supplement 1B*). Mice receiving APAP plus X209, HyperIL6, or HyperIL6+ iSTAT3 were equally protected from liver damage with lower ALT/AST, higher GSH, and greater expression of PCNA/Cyclin D1/Ki67, while having reduced centrilobular necrosis. At the signaling level, mice receiving APAP plus X209, HyperIL6, or HyperIL6+ iSTAT3 had similarly reduced ERK and JNK signaling. Only mice with HyperIL6 alone had increased STAT3 phosphorylation that was unrelated to the phenotypes studied here (*Figure 3G–K*, *Figure 3—figure supplement 1B*). Thus the beneficial effects of HyperIL6 on hepatoprotection and regeneration are STAT3 independent at this later time point of assessment.

Our hypothesis (*Figure 1H*), and data (*Figures 1–3*), propose that the beneficial effects of HyperIL6 are due to its inhibition of IL11 signaling. To test this specifically, we injected recombinant mouse IL11 (rmIL11) to mice with HyperIl6 expression± S3I-201 (*Figure 3L*). Injection of rmIL11 to mice (6 hr) resulted in elevated ALT/AST levels and activation of ERK and JNK, as expected (*Figure 3M,N*, *Figure 3—figure supplement 1C*; *Widjaja et al., 2021*). Following rmIL11 injection, mice expressing HyperIL6 had elevated STAT3 phosphorylation, lower ALT/AST levels and lesser activation of ERK and JNK, as compared to controls. Administration of S3I-201 to HyperIL6

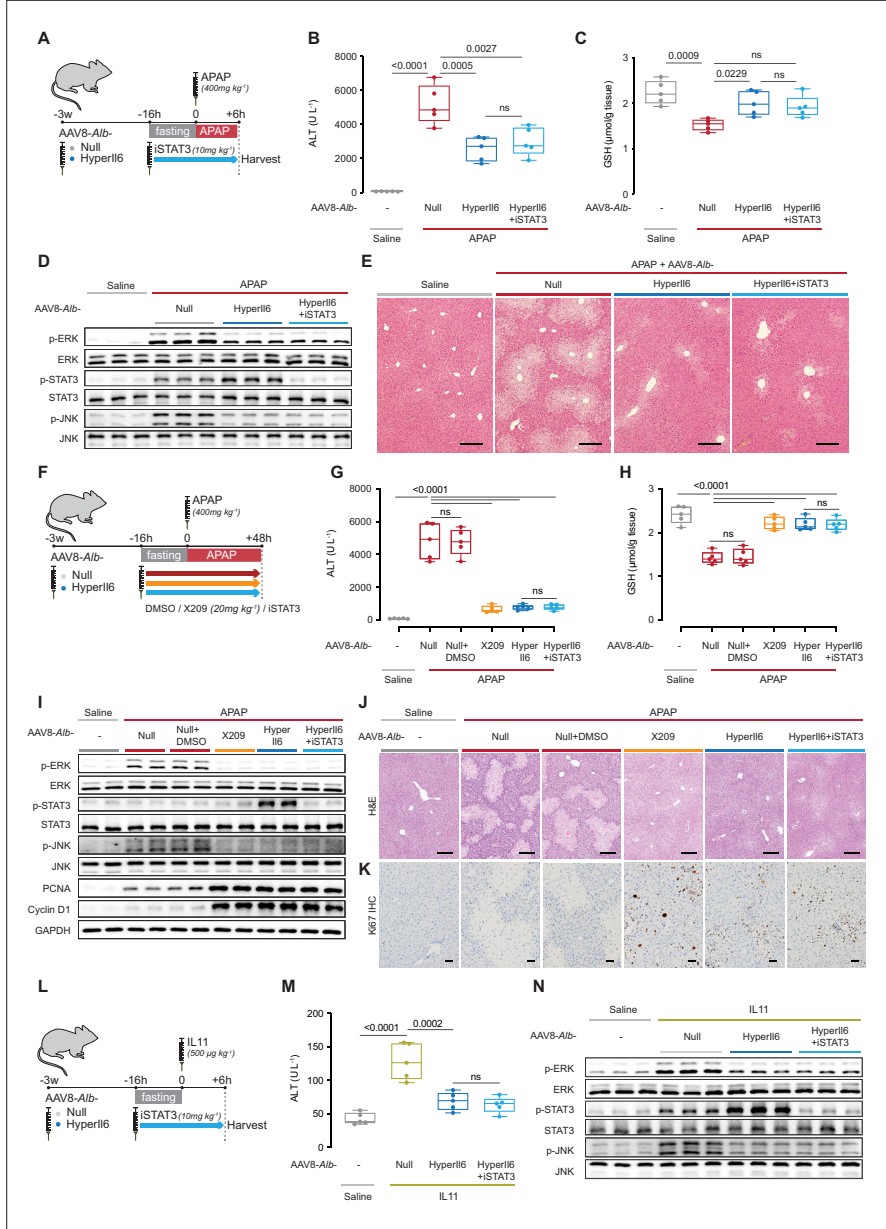

**Figure 3.** Hepatocyte-specific HyperIl6 expression reduces APAP- or IL11-induced liver injury independent of STAT3 activation. (**A**) Schematic of APAP-injected mice with hepatocyte-specific expression of HyperIl6± iSTAT3 administration. Three weeks following AAV8-*Alb*-Null or AAV8-*Alb*-HyperIl6 virus injection, mice were injected with APAP (400 mg/kg); iSTAT3 (S3I-201, 10 mg/kg) was administered at the beginning of fasting period, 16 hr prior to APAP injection; control mice received saline injection; mice were harvested 6 hr post-saline or APAP injection. (**B**) Serum ALT levels, (**C**) hepatic GSH levels, (**D**) western blots showing hepatic phospho-ERK, ERK, phospho-STAT3, STAT3, phospho-JNK, and JNK, and (**E**) representative H&E-stained liver images (scale bars, 50 μm) for experiments shown in (**A**). (**F**) Schematic of APAP-injected mice with hepatocyte-specific expression of HyperIl6 or DMSO/X209/iSTAT3 administration. Three weeks following AAV8-*Alb*-Null or AAV8-*Alb*-HyperIl6 virus injection, mice were injected with APAP (400 mg/kg). DMSO (0.2 ml/kg), X209 (20 mg/kg), or iSTAT3 (10 mg/kg) were administered at the beginning of fasting period, 16 hr prior to APAP injection; control mice received saline injection; mice were harvested 48 hr post-saline or APAP injection. (**G**) Serum ALT levels, (**H**) hepatic GSH levels, and (**I**) western blots showing hepatic levels of phospho-ERK, ERK, phospho-STAT3, STAT3, phospho-JNK, JNK, PCNA, Cyclin D1, and GAPDH as internal control, (**J**) representative H&E-stained liver images (scale bars, 50 μm), (**K**) immunohistochemistry staining of Ki67 in the livers of mice (scale bars, 50 μm) for experiments shown in (**F**). (**L**) Schematic of rmIL11-injected mice with hepatocyte-specific expression of HyperIl6 ± iSTAT3 administration. Mice were injected with rmIL11 (500 μg/kg), 3 weeks following AAV8-*Alb*-Null or AAV8-*Alb*-HyperIl6 virus injection;

*Figure 3 continued on next page*

*Figure 3 continued*

iSTAT3 (10 mg/kg) was administered at the beginning of fasting period, 16 hr prior to rmIL11 injection; control mice received saline injection; mice were harvested 6 hr post-saline or IL11 injection. (**M**) Serum ALT levels and (**N**) western blots showing hepatic ERK, STAT3, and JNK activation status for experiments shown in (**L**). (**B–C, G–H, M**) N = 5 mice/group; data are shown as box-and-whisker with median (middle line), 25th–75th percentiles (box), and min–max values (whiskers), one-way ANOVA with Tukey's correction.

The online version of this article includes the following figure supplement(s) for figure 3:

**Source data 1.** Raw data, western blot quantification, necrotic area (%) of H&E-stained liver images, and positive cell counts of Ki67-stained liver images for panels B–E, G–K, M, N.

**Source data 2.** Western blot images (original and annotated) for panels D, I, N.

**Figure supplement 1.** Hepatocyte-specific HyperIl6 expression reduces APAP- or IL11-induced liver injury independent of STAT3 activation.

**Figure supplement 1—source data 1.** Raw data for panels A–C.

---

expressing mice reduced STAT3 activity to baseline but had no effect on its beneficial outcomes at any level of assessment. (*Figure 3M,N*, *Figure 3—figure supplement 1C*).

## HyperIL6 has no effect on APAP-induced liver injury in mice deleted for *Il11*

If the protective effects of HyperIL6 are due to its inhibition of IL11 signaling, then HyperIL6 should be ineffective in APAP injury in the absence of IL11. Thus we studied the impact of HyperIL6 on APAP-induced liver injury in mice globally deleted for *Il11* (*Il11⁻/⁻*) (*Figure 4A*; *Ng et al., 2021*).

APAP dosing resulted in increased IL11 expression in the injured livers of wild-type (WT) mice that was, as expected, absent in *Il11⁻/⁻* mice (*Figure 4B*). Following APAP, as compared to WT controls, expression of HyperIL6 in WT mice was associated with lesser liver damage and a molecular signature of regeneration (*Figure 4B–F*). As compared to WT mice receiving APAP, *Il11⁻/⁻* mice dosed with APAP had reduced ALT, AST, and centrilobular necrosis, higher GSH levels along with increased Ki67, PCNA, and Cyclin D1 expression (*Figure 4B–F*, *Figure 4—figure supplement 1*). Thus lack of IL11 signaling due to genetic deletion if *Il11* stimulates regeneration. Notably, expression of HyperIL6 had no additive effect on hepatoprotection or liver regeneration in *Il11⁻/⁻* mice.

At the signaling level, APAP-related ERK and JNK activation were reduced in both HyperIL6-expressing WT mice and in *Il11⁻/⁻* mice in the absence of HyperIL6 (*Figure 4G*). While HyperIL6 expression robustly increased STAT3 phosphorylation in both WT and *Il11⁻/⁻* mice, this activity was unrelated to liver protection or regeneration (*Figure 4B–G*).

## Conclusion

For almost three decades now, IL6 signaling, in particular HyperIL6 activation of STAT3, has been thought to promote liver regeneration (*Cressman et al., 1996*; *James et al., 2003*). While some early reports questioned this assertion (*Sakamoto et al., 1999*), it is now generally accepted (*Schmidt-Arras and Rose-John, 2016*). Here we show that HyperIL6-mediated inhibition of IL11 signaling (NOX4, ERK, and JNK; *Widjaja et al., 2021*; *Widjaja et al., 2020*) in APAP-injured hepatocytes, latent until now, is the dominant mechanism underlying the pro-regenerative effects of HyperIL6 in the damaged liver. We postulate that competition of HyperIL6 with IL11:IL11RA complexes for binding to IL6ST could explain why injection of HyperIL6, but not IL6 itself, promotes liver regeneration (*Nechemia-Arbely et al., 2011*). Our study suggests that caution is needed when interpreting assumed gain-of-function, on-target effects of synthetic IL6ST-interacting molecules such as HyperIL6, Nₜ-3N (*Nishina et al., 2012*) or the recently described IC7Fc fusion molecule (*Findeisen et al., 2019*). IL6ST-related ligand, interacting alpha receptor, and signaling pleiotropy is large, and mechanism of effect is hard to decipher using overexpression of synthetic and alien factors. We end by suggesting IL11 instead of IL6 as a focus for regenerative studies of the liver and perhaps nerves (*Fischer, 2017*; *Leibinger et al., 2021*) and kidney (*Nechemia-Arbely et al., 2008*). With anti-IL11 therapies advancing toward the clinic, this provides interesting opportunities.

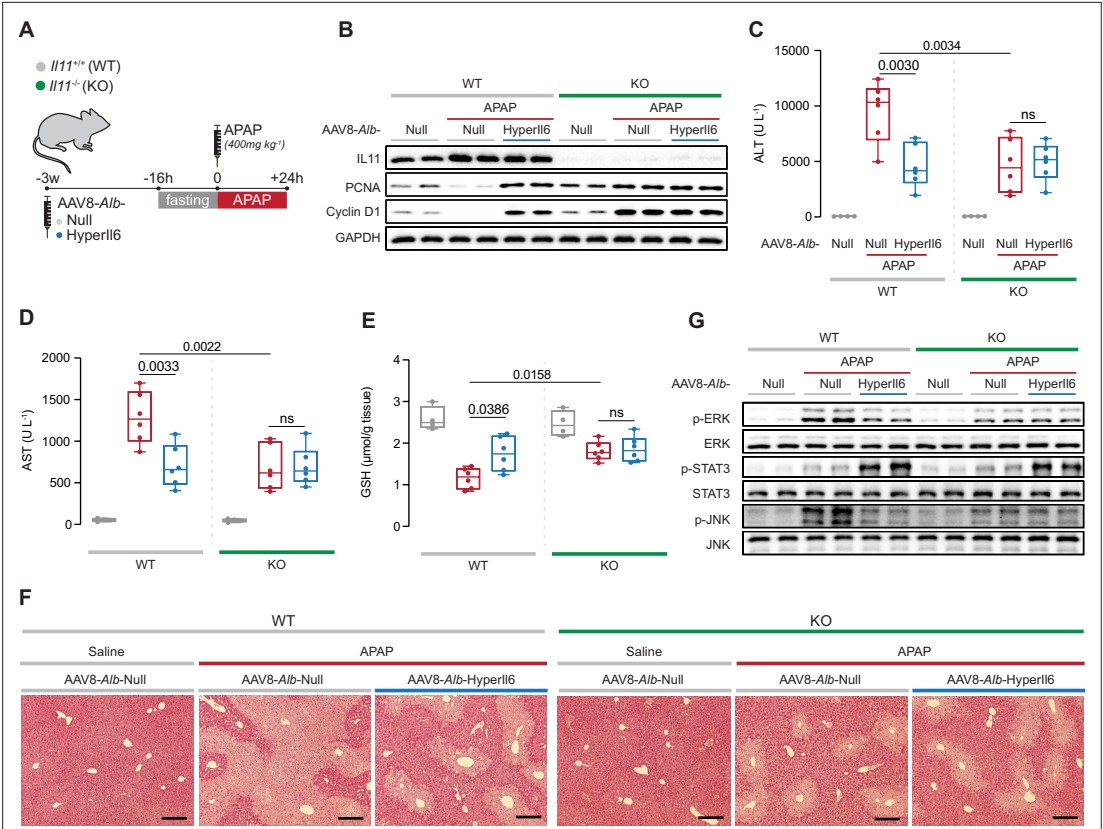

**Figure 4.** *Il11⁻/⁻* mice exhibit spontaneous liver regeneration following APAP injury and HyperIL6 has no beneficial effect in this strain. (**A**) Schematic of APAP injury in *Il11⁻/⁻* and *Il11⁺/⁺* mice (control) with hepatocyte-specific expression of HyperIl6. Three weeks following AAV8-*Alb*-Null or AAV8-*Alb*-HyperIl6 virus injection, overnight-fasted *Il11⁺/⁺* and *Il11⁻/⁻* mice were injected with saline or APAP (400 mg/kg); mice were harvested 24 hr post-saline or APAP injection. (**B**) Western blots showing hepatic levels of IL11, PCNA, Cyclin D1, and GAPDH as internal control. (**C**) Serum ALT levels. (**D**) Serum AST levels. (**E**) Hepatic GSH levels. (**F**) Representative H&E-stained liver images (scale bars, 50 μm). (**G**) Western blots showing hepatic phospho-ERK, ERK, phospho-STAT3, STAT3, phospho-JNK, and JNK. (**C–E**) Saline (n = 4 mice/group), APAP (n = 6 mice/group); data are shown as box-and-whisker with median (middle line), 25th–75th percentiles (box), and min–max values (whiskers), two-way ANOVA with Sidak's correction.

The online version of this article includes the following figure supplement(s) for figure 4:

**Source data 1.** Raw data, western blot quantification, and necrotic area (%) of H&E-stained liver images for panels B–F.

**Source data 2.** Western blot images (original and annotated) for panels B and G.

**Figure supplement 1.** Hepatocyte-specific HyperIl6 expression has no effect on hepatocyte regeneration in *Il11⁻/⁻* mice.

**Figure supplement 1—source data 1.** Positive cell counts of Ki67-stained liver images.

# Materials and methods

## Key resources table

| Reagent type (species) or resource | Designation | Source or reference | Identifiers | Additional information |
|---|---|---|---|---|
| Strain, strain background (mouse) | *Il11⁻/⁻* mice *C57BL/6* J | PMID:34239012 | | Crispr/Cas9 technique was used to knock out the *Il11* gene. |
| Cell line (*Homo sapiens*) | Hepatocytes | ScienCell | Cat# 5,200 | Primary cell line |
| Antibody | Phospho-AKT (Rabbit monoclonal) | CST | Cat# 4060; RRID:AB_2315049 | WB (1:1000) |
| Antibody | AKT (Rabbit monoclonal) | CST | Cat# 4691; RRID:AB_915783 | WB (1:1000) |

*Continued on next page*

*Continued*

| Reagent type (species) or resource | Designation | Source or reference | Identifiers | Additional information |
|---|---|---|---|---|
| Antibody | Cyclin D1 (Rabbit monoclonal) | CST | Cat# 55506; RRID:AB_2827374 | WB (1:1000) |
| Antibody | phospho-ERK1/2 (Rabbit monoclonal) | CST | Cat# 4370; RRID: AB_2315112 | WB (1:1000) |
| Antibody | ERK1/2 (Rabbit monoclonal) | CST | Cat# 4695; RRID: AB_390779 | WB (1:1000) |
| Antibody | GAPDH (Rabbit monoclonal) | CST | Cat# 2118; RRID: AB_561053 | WB (1:1000) |
| Antibody | IgG (11E10; mouse IgM isotype control) | PMID:31078624; Aldevron | | In vivo neutralizing experiment (20 mg/kg) |
| Antibody | IL6 (Goat polyclonal) | R&D systems | Cat# AF506; RRID:AB_355398 | WB (1 µg/ml) |
| Antibody | IL6ST (extracellular; Rabbit polyclonal) | Thermo Fisher | Cat# PA5-77476; RRID:AB_2735869 | WB (1:1000) |
| Antibody | IL6ST (Mouse monoclonal) | R&D systems | Cat# MAB628; RRID:AB_2125962 | In vitro neutralizing experiment (2 µg/ml) |
| Antibody | IL11 (X203; mouse monoclonal) | PMID:31078624; Aldevron | | WB (1 µg/ml) |
| Antibody | IL11RA (X209; mouse monoclonal) | PMID:31078624; Aldevron | | In vivo neutralizing experiment (20 mg/kg) |
| Antibody | p-JNK (Rabbit monoclonal) | CST | Cat# 4668; RRID:AB_823588 | WB (1:1000) |
| Antibody | JNK (Rabbit polyclonal) | CST | Cat# 9252; RRID:AB_2250373 | WB (1:1000) |
| Antibody | Ki67 (Rabbit monoclonal) | Abcam | Cat# ab16667; RRID:AB_302459 | IHC (1:200) |
| Antibody | NOX4 (Rabbit monoclonal) | Thermo Fisher | Cat# MA5-32090; RRID: AB_2809383 | WB (1:1000) |
| Antibody | PCNA (Rabbit monoclonal) | CST | Cat# 13110; RRID:AB_2636979 | WB (1:1000) |
| Antibody | phospho-STAT3 (Mouse monoclonal) | CST | Cat# 4113; RRID: AB_2198588 | WB (1:1000) |
| Antibody | STAT3 (Rabbit monoclonal) | CST | Cat# 4904; RRID: AB_331269 | WB (1:1000) |
| Antibody | anti-mouse HRP (Horse polyclonal) | CST | Cat# 7076; RRID:AB_330924 | WB (1:2000) |
| Antibody | anti-rabbit HRP (Goat polyclonal) | CST | Cat# 7074; RRID:AB_2099233 | WB (1:2000); IHC (1:200) |
| Antibody | anti-rat HRP (Goat polyclonal) | Abcam | Cat# ab97057; RRID:AB_10680316 | WB (1:2000) |
| Recombinant DNA reagent | AAV8-*Alb*-HyperIl6 | This paper; Vector Biolabs | | AAV8 vector expressing mouse HyperIl6. See Materials and methods, AAV8 vectors. |
| Recombinant DNA reagent | AAV8-*Alb*-sIl6st | PMID:33397952; Vector Biolabs | | AAV8 vector expressing mouse sIL6ST. |
| Peptide, recombinant protein | rhIL11 | PMID:29160304; Genscript | Cat# Z03108 | UniProtKB: P20809 |
| Peptide, recombinant protein | rmIL11 | PMID:29160304; Genscript | Cat# Z03052 | UniProtKB: P47873 |
| Peptide, recombinant protein | rh-HyperIL6 | R&D systems | Cat# 8954 SR | Human IL6R:IL6 fusion protein |

*Continued on next page*

*Continued*

| Reagent type (species) or resource | Designation | Source or reference | Identifiers | Additional information |
|---|---|---|---|---|
| Peptide, recombinant protein | rm-HyperIL6 | R&D systems | Cat# 9038 SR | Mouse IL6R:IL6 fusion protein |
| Peptide, recombinant protein | soluble IL6ST Fc | R&D systems | Cat# 671-GP-100 | |
| Commercial assay or kit | ALT Activity Assay Kit | Abcam | Cat# ab105134 | |
| Commercial assay or kit | AST Activity Assay Kit | Abcam | Cat# ab105135 | |
| Commercial assay or kit | Glutathione Colorimetric Detection Kit | Thermo Fisher | Cat# EIAGSHC | |
| Chemical compound, drug | APAP | Sigma | Cat# A3035 | |
| Chemical compound, drug | DMSO | Sigma | Cat# D2650 | |
| Chemical compound, drug | iSTAT3 (S3I-201) | Sigma | Cat# SML0330 | |
| Software, algorithm | GraphPad Prism | GraphPad Prism | RRID:SCR_002798 | Version 6.07 |
| Software, algorithm | ImageJ | ImageJ | RRID:SCR_003070 | |
| Other | Hoechst 33,342 | Thermo Fisher | Cat# 62,249 | Operetta high-throughput phenotyping assay |
| Other | DRAQ7 | Thermo Fisher | Cat# D15106 | Operetta high-throughput phenotyping assay |
| Other | DCFDA | Abcam | Cat# ab113851 | ROS stain |
| Other | BOND Polymer Refine Detection Kit | Leica | Cat# DS9800; RRID:AB_2891238 | IHC stain |

## AAV8 vectors

All AAV8 vectors used in this study were synthesized by Vector Biolabs. AAV8 vector carrying mouse HyperIl6 cDNA driven by *Alb* promoter is referred to as *AAV8-Alb*-HyperIl6, which was constructed using the cDNA sequences of mouse IL6/IL6R alpha fusion protein (9038 SR, R&D systems). AAV8-*Alb*-Null vector was used as vector control.

## Cell culture

Primary human hepatocytes (5200, ScienCell) were maintained in hepatocyte medium (5201, Scien-Cell) supplemented with 2 % fetal bovine serum, 1 % penicillin-streptomycin at 37 °C and 5 % $CO_2$. Hepatocytes were serum-starved overnight unless otherwise specified in the methods prior to 24 hr stimulation with different doses of various recombinant proteins as outlined in the main text and/or figure legends. All experiments were carried out at low cell passage (< P3).

## Operetta high-throughput phenotyping assay

Primary human hepatocytes were seeded in 96-well black CellCarrier plates (PerkinElmer) at a density of $5 \times 10^3$ cells per well. Following stimulations, cells were incubated 1 hr with 1 μg/ml Hoechst 33,342 (62249, Thermo Fisher Scientific) and DRAQ7 (D15106, Thermo Fisher Scientific) in serum-free basal medium. Each condition was imaged from triplicated wells and a minimum of 23 fields/well using Operetta high-content imaging system 1483 (PerkinElmer). Live and dead cells were quantified using Harmony v3.5.2 (PerkinElmer).

## ROS detection

Primary human hepatocytes were seeded on eight-well chamber slides ($1.5 \times 10^4$ cells/well). For this experiment, cells were not serum-starved prior to treatment. Twenty-four hours following stimulation,

cells were washed, incubated with 25 µM of DCFDA solution (ab113851, abcam) for 45 min at 37 °C in the dark, and rinsed with the dilution buffer according to the manufacturer's protocol. Live cells with positive DCF staining were imaged with a filter set appropriate for fluorescein (FITC) using a fluorescence microscope (Leica).

### Animal models

Animal procedures were approved and conducted in accordance with the SingHealth Institutional Animal Care and Use Committee (IACUC). All mice were housed in temperatures of 21–24°C with 40–70% humidity on a 12 hr light/12 hr dark cycle and provided food and water ad libitum, except in the fasting period, during which only water was provided ad libitum.

### Mouse models of APAP

Prior to APAP, 9–12 weeks old male mice were fasted overnight. Mice were given APAP (400 mg/kg) by intraperitoneal (IP) administration and euthanized 6 hr, 24 hr, or 48 hr post-APAP, as outlined in the main text or figure legends.

### In vivo administration of Rh-HyperIL6, Rm-HyperIL6, or rmIL11

rh-HyperIL6, rm-HyperIL6, or rmIL11 were administered *via* IP injection at a concentration of 500 µg/kg.

### In vivo expression of HyperIl6 or sIl6st

Six to 8 weeks old male C57BL/6NTac mice (InVivos, Singapore) were injected with $4 \times 10^{11}$ gc AAV8-*Alb*-HyperIl6 or AAV8-*Alb*-s*Il6st* virus to induce hepatocyte-specific expression of HyperIl6 or s*Il6st*; control mice were injected with $4 \times 10^{11}$ gc AAV8-*Alb*-Null virus. Three weeks following virus administration, mice were given IP administration of APAP and euthanized at the time point outlined in the main text or figure legends.

### In vivo administration of anti-IL11RA (X209) or iSTAT3 (S3I-201)

C57BL/6NTac male mice were IP administered anti-IL11RA (X209, 20 mg/kg), IgG isotype control (11E10, 20 mg/kg), or iSTAT3 (S3I-201, 10 mg/kg) at the beginning of fasting period.

### Il11−/− mice

Mice lacking functional alleles for Il11 (*Il11−/−*), in which Crispr/Cas9 technique was used to knock out the *Il11* gene (ENSMUST00000094892.11), were generated and validated previously (**Ng et al., 2021**). Six to 8 weeks old male *Il11−/−* mice and their WT littermates (*Il11+/+*) were injected with $4 \times 10^{11}$ gc AAV8-*Alb*-HyperIl6 virus to induce hepatocyte-specific expression of HyperIl6; control mice were injected with $4 \times 10^{11}$ gc AAV8-*Alb*-Null virus. Three weeks following virus administration, mice were given IP administration of APAP and euthanized 24 hr post-APAP.

### Colorimetric assays

The levels of ALT or AST in mouse serum and hepatocyte supernatant were measured using ALT (ab105134, Abcam) or AST (ab105135, Abcam) Activity Assay Kits. Liver GSH measurements were performed using the Glutathione Colorimetric Detection Kit (EIAGSHC, Thermo Fisher). All colorimetric assays were performed according to the manufacturer's protocol.

### Immunoblotting

Western blots were carried out from hepatocyte and liver tissue lysates. Hepatocytes and tissues were homogenized in radioimmunoprecipitation assay (RIPA) buffer containing protease and phosphatase inhibitors (Thermo Fisher), followed by centrifugation to clear the lysate. Protein concentrations were determined by Bradford assay (Bio-Rad). Equal amounts of protein lysates were separated by SDS–PAGE, transferred to PVDF membrane, and subjected to immunoblot analysis for the indicated primary antibodies. Proteins were visualized using the ECL detection system (Pierce) with the appropriate secondary antibodies.

## Surface plasmon resonance

Surface plasmon resonance (SPR) measurements were performed on a BIAcore T200 (GE Healthcare) at 25 °C. Buffers were degassed and filter-sterilized through 0.2 µm filters prior to use. IL6ST was immobilized onto a carboxymethylated dextran (CM5) sensor chip using standard amine coupling chemistry. For kinetic analysis, a concentration series (0.39 nM to 120 nM) of IL6, HyperIL11, or HyperIL6 was injected over the IL6ST and reference surfaces at a flow rate of 30 µl/min. All the analytes were dissolved in HBS-EP+ (BR100669, GE Healthcare) containing 1 mg/ml BSA. The association and dissociation were measured for 210 s and 300 s, respectively. After each analyte injection, the surface was regenerated by two times injection of Glycine-HCl (10 mM, pH 1.5), followed by a 5 min stabilization period. All sensorgrams were aligned and double-referenced. Affinity and kinetic constants were determined by fitting the corrected sensorgrams with the 1:1 Langmuir model using BIAevaluation v3.0 software (GE Healthcare). The equilibrium binding constant $K_D$ was determined by the ratio of the binding rate constants $k_d/k_a$.

## Histology

### Hematoxylin and eosin staining

Livers were fixed for 48 hr at room temperature in 10 % neutral-buffered formalin (NBF), dehydrated, embedded in paraffin blocks, and sectioned at 7 µm. Sections were stained with hematoxylin and eosineosin (H&E) according to standard protocol and examined by light microscopy.

### Immuno-histochemistry staining

Livers were processed as mentioned above (H&E staining section). Following dewaxing and antigen retrieval, liver sections were stained with a BOND Polymer Refine Detection Kit (DS9800, Leica) by BOND-III Automated IHC/ISH Stainer (Leica). Ki67 staining was examined by light microscopy.

## Statistical analysis

Statistical analyses were performed using GraphPad Prism software (version 6.07). For comparisons between more than two conditions, one-way ANOVA with Dunnett's correction (when several conditions were compared to one condition) or Tukey's correction (when several conditions were compared to each other) were used. Comparison analysis for several conditions from two different groups was performed by two-way ANOVA and corrected with Sidak's multiple comparisons when the means were compared to each other. The criterion for statistical significance was $p < 0.05$.

## Acknowledgements

This research was supported by the National Medical Research Council (NMRC), Singapore STaR awards (NMRC/STaR/0029/2017), NMRC Centre Grant to the NHCS, MOH-CIRG18nov-0002, MRC-LMS (UK), Tanoto Foundation to SAC. AAW is supported by NMRC/OFYIRG/0053/2017. The authors would like to acknowledge the technical support of BL George and J Tan.

## Additional information

### Competing interests

Sebastian Schafer: is a co-inventor of the patent applications: WO/2017/103108 (Treatment of Fibrosis), WO/2018/109174 (IL11 Antibodies), WO/2018/109170 (IL11RA Antibodies), and US 2020/0262910 (Treatment of Hepatotoxicity). Is a co-founder and shareholder of Enleofen Bio PTE LTD.. Anissa A Widjaja: is a co-inventor of the patent application: US 2020/0262910 (Treatment of Hepatotoxicity).. Stuart A Cook: is a co-inventor of the patent applications: WO/2017/103108 (TREATMENT OF FIBROSIS), WO/2018/109174 (IL11 ANTIBODIES), WO/2018/109170 (IL11RA ANTIBODIES), and US 2020/0262910 (Treatment of Hepatotoxicity). S.A.C. is a co-founder and shareholder of Enleofen Bio PTE LTD.. The other authors declare that no competing interests exist.

## Funding

| Funder | Grant reference number | Author |
|---|---|---|
| National Medical Research Council | NMRC/STaR/0029/2017 | Stuart A Cook |
| National Medical Research Council | NMRC Centre Grant to the NHCS | Stuart A Cook |
| National Medical Research Council | MOH-CIRG18nov-0002 | Stuart A Cook |
| Medical Research Council | MRC-LMS | Stuart A Cook |
| Goh Foundation | | Stuart A Cook |
| Tanoto Foundation | | Stuart A Cook |
| National Medical Research Council | NMRC/OFYIRG/0053/2017 | Anissa A Widjaja |

The funders had no role in study design, data collection and interpretation, or the decision to submit the work for publication.

## Author contributions

Jinrui Dong, Formal analysis, Investigation, Project administration, Validation, Writing – original draft; Sivakumar Viswanathan, Formal analysis, Investigation; Eleonora Adami, Visualization; Sebastian Schafer, Supervision; Fathima F Kuthubudeen, Investigation; Anissa A Widjaja, Methodology, Project administration, Supervision, Writing – review and editing; Stuart A Cook, Conceptualization, Funding acquisition, Methodology, Supervision, Writing – original draft, Writing – review and editing

## Author ORCIDs

Anissa A Widjaja (ID) http://orcid.org/0000-0001-9404-7608
Stuart A Cook (ID) http://orcid.org/0000-0001-6628-194X

## Ethics

Animal studies were carried out in compliance with the recommendations in the Guidelines on the Care and Use of Animals for Scientific Purposes of the National Advisory Committee for Laboratory Animal Research (NACLAR). All experimental procedures were approved (SHS/2014/0925 and SHS/2019/1482) and conducted in accordance with the SingHealth Institutional Animal Care and Use Committee.

## Decision letter and Author response

Decision letter https://doi.org/10.7554/eLife.68843.sa1
Author response https://doi.org/10.7554/eLife.68843.sa2

# Additional files

## Supplementary files
• Transparent reporting form

## Data availability

All data generated or analysed during this study are included in the manuscript and supporting files. Source data are provided with this paper.

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
