## [Decision Letter]

**Acceptance summary:**

This paper will have a high impact on the field, as it identifies a new mechanism for the effects of IL6 on liver regeneration and response to injury. The techniques and methodology are also sound and multiple approaches are utilized to answer the main question of the paper.

**Decision letter after peer review:**

Thank you for submitting your article "Overturning the paradigm that IL6 signaling drives liver regrowth while shining light on a new therapeutic target for regenerative medicine" for consideration by *eLife*. Your article has been reviewed by 3 peer reviewers, including Hossein Ardehali as the Reviewing Editor and Reviewer #3, and the evaluation has been overseen by Mone Zaidi as the Senior Editor.

Essential revisions (for the authors):

1) This manuscript can be significantly improved by limiting conclusions of the study to APAP-induced liver injury rather than broad and general conclusions about liver regeneration. Similarly title of the manuscript can be changed to be specific about the APAP liver injury model to avoid misinterpretation.

2) In order to properly study liver regeneration, important time points for peak proliferation such as 48 hr post-APAP should be examined. PCNA western blot should be substantiated with conclusive staining (e.g. Ki67).

3) The findings that protective effects of HyperIL6 against APAP toxicity are Il11 dependent can be substantiated by studies examining if Il11 downstream signaling pathway is activated during APAP toxicity and if it is inhibited by hyperIL6 treatment in vivo. Figure 4 shows effects of HyperIL6 on APAP toxicity in Il11 KO mice, which can be strengthened by examining if Il11 overexpression can overcome hyperIl6 competition and reverse protective effect of HyperIL6 on APAP hepatotoxicity in vivo. Further similar to data in Figure 1i on IL11 toxicity, manuscript can be bolstered by data showing that protective effect of HyperIL6 on APAP toxicity can by titrated away by the addition of soluble gp130.

4) Reference for generation and validation of important mouse strain (Il11 ko) is missing, which should be added. Further, background strain of these mice should be mentioned.

5) The authors assessed ROS production in response to APAP and other reagents in Figure 1. However, no mitochondrial studies are performed. This reviewer is not asking for the mechanism of cellular protection, and agrees with the authors that the paper contains sufficient novelty, but including mitochondrial studies would strengthen the data.

*Reviewer Recommendations for the authors:*

The paper will have a high impact on the field, as it identifies a new mechanism for the effects of IL6 on liver regeneration. The techniques and methodology are also sound and multiple approaches are utilized to answer the main question of the paper.

1) The major limitation of this study is that the broad conclusion about "overturning the paradigm that Il6 promotes liver regeneration" is not convincingly supported by the presented data. The manuscript is mostly focused on the injury phase of APAP hepatotoxicity (6 and 24 hr post-APAP) and provides limited information on the peak regeneration phase (i.e. later time points – e.g. 48 and 72 hr post-APAP). Further, this is complicated by the fact that liver regeneration in APAP model is a compensatory phenomenon secondary to liver injury. Thus, any effect on liver injury (as observed in this manuscript) will indirectly alter liver regeneration response. Thus, it is hard to interpret about direct role of Il6 in liver regeneration in the current study. Overall, it is incorrect to interpret in general about role of Il6 in regeneration based on the limited data in the APAP model presented in this study. Cleaner model such as partial hepatectomy can be utilized for such interpretations. Further, it will be incorrect to interpret physiological role of Il6 solely based on experiments utilizing HyperIL6 overexpression. In this regard authors should discuss important previous study by James et al. utilizing Il6 KO mice in APAP model which showed that despite no effect on initial APAP-induced liver injury, Il6 KO mice displayed reduced liver regeneration and slower recovery (PMID: 13679052).

2) In Figure 3, in order to demonstrate that protective effects of HyperIL6 are independent of STAT3, authors used STAT3 inhibitor (iSTAT3). Authors investigated only single time point (6 hour post-APAP), which does not provide information if stimulation of liver regeneration by hyperIL6 is independent of STAT3. In order to study liver regeneration, authors should study later time points such as 48 hr post-APAP or at least 24 hr time point as done for other figures in the paper. Further, it is not clear from the manuscript if vehicle control group for iSTAT3 was utilized in this experiment. From in-vitro experiments presented earlier in the manuscript, I assume DMSO was the vehicle for iSTAT3. If that's the case, proper vehicle control is required as DMSO is known to inhibit Cyp2e1 and APAP metabolism to toxic metabolite and thus show protective effects against APAP toxicity.

3) This study does provide interesting data on the role of IL11 in APAP-induced liver injury but with limited mechanistic details. Based on these findings, more comprehensive and mechanistic studies in future will be helpful to delineate how IL11 regulate APAP-induced liver injury. It will also be important to rule out any impact of Il11 deletion on metabolic activation of APAP.

---

## [Author Response]

Essential revisions:1) This manuscript can be significantly improved by limiting conclusions of the study to APAP-induced liver injury rather than broad and general conclusions about liver regeneration. Similarly title of the manuscript can be changed to be specific about the APAP liver injury model to avoid misinterpretation.

We agree and have amended the manuscript accordingly. The title has been specifically adjusted, mention of IL6 itself removed. The new title reads: “The pro-regenerative effects of HyperIL6 in drug induced liver injury are unexpectedly due to competitive inhibition of IL11 signaling”.

2) In order to properly study liver regeneration, important time points for peak proliferation such as 48 hr post-APAP should be examined. PCNA western blot should be substantiated with conclusive staining (e.g. Ki67).

We agree that liver regeneration peaks some 48 hours post-injury. The molecular and cellular markers of regeneration are apparent earlier (e.g. at 24 hours), as we showed, but it is true that effects may be more pronounced at later time points (e.g. 48 hours). To extend our findings further, we have carried out new experiments at a 48 hour time point and show the data in the revision. In addition we have now added staining for Ki67, as another marker of proliferation, as requested. Staining for Ki67 mirrors the previous data, with a marked increase following APAP seen with anti-IL11RA (X209), HyperIL6 or HyperIL6+iSTAT3 (i.e., the HyperIL6 effect remains STAT3-independent at 48 h). See revised Figure 3 and revised Figure 2—figure supplement 1.

3) The findings that protective effects of HyperIL6 against APAP toxicity are Il11 dependent can be substantiated by studies examining if Il11 downstream signaling pathway is activated during APAP toxicity and if it is inhibited by hyperIL6 treatment in vivo.

During the revision of the current manuscript we published a study on the pathomechanisms underlying IL11 effect in APAP-induced liver injury (Widjaja et al. 2021). In brief, the maladaptive signaling pathways downstream of IL11 following APAP injury are ERK and *JNK* mediated and related, in part, to IL11-dependent NOX4 upregulation. In the current study, we show across multiple figure panels that these pathways are consistently inhibited by HyperIL6 (in a STAT3-independent fashion) both in vitro and in vivo following either APAP injury or IL11 administration (revised Figures 1G and J; 2E and L; 3D, I and N).

Figure 4 shows effects of HyperIL6 on APAP toxicity in Il11 KO mice, which can be strengthened by examining if Il11 overexpression can overcome hyperIl6 competition and reverse protective effect of HyperIL6 on APAP hepatotoxicity in vivo.

We believe our experimental approach of gain-of-function (HyperIL6 expression) on the background of loss-of-function (IL11 KO) in the context of APAP is the most informative experimental design to address our hypothesis, in this context. We do not think the proposed approach of gain-of-function (HyperIL6 expression) combined with further gain-of-function (IL11 expression) in APAP would be insightful. The liver produces vast amounts of IL11 when damaged by APAP, indeed systemic levels go up from low pM amounts to up to 4ng/ml, and it is unlikely that systemically administered recombinant IL11 would have any effect on top of already extremely high endogenous IL11 levels.

Further similar to data in Figure 1i on IL11 toxicity, manuscript can be bolstered by data showing that protective effect of HyperIL6 on APAP toxicity can by titrated away by the addition of soluble gp130.

We thank the reviewer for the suggestion. In response, we have performed new experiments to examine whether or not the protective effect of HyperIL6 on APAP toxicity can be titrated away by the addition of soluble gp130 (sIL6ST). We also went one step further to determine if effects are influenced by HyperIL6-induced STAT3 activation, or not. The data from these experiments, Western blotting and GSH assessment, are presented in the revised manuscript in revised Figure 1—figure supplement 1. In short, we show that the protective effects of HyperIL6 can indeed be titrated away by sIL6ST. We also show, once again, that STAT3 activation is unrelated to the protective effects of HyperIL6.

j4) Reference for generation and validation of important mouse strain (Il11 ko) is missing, which should be added. Further, background strain of these mice should be mentioned.

The manuscript describing the new *Il11* knockout has now been published and is referenced fully in the revised manuscript, as it is here (Ng et al., 2021).

5) The authors assessed ROS production in response to APAP and other reagents in Figure 1. However, no mitochondrial studies are performed. This reviewer is not asking for the mechanism of cellular protection, and agrees with the authors that the paper contains sufficient novelty, but including mitochondrial studies would strengthen the data.

We thank the Reviewer for his/her interest in a deeper understanding of the pathomechanisms. While working on this revision for *eLife* we have published a paper that was on BioRxiv, and referenced in the original manuscript, which has now been published in *Science Translational Medicine (Widjaja et al., 2021)*. This new study shows in detail that IL11 drives NOX4 to produce ROS downstream of NAPQI-induced mitochondrial damage and that ERK/*JNK* signaling is central to the maladaptive effects of autocrine IL11 signaling in APAP-injured hepatocytes. We mention this mechanism in the revision and cite the new manuscript. In the *Science Translational Medicine* paper we did not present data on mitochondrial function but are happy to share here – for the Reviewer’s information – data showing that inhibition of IL11 signaling with an anti-IL11RA antibody preserves mitochondrial function in APAP-injured primary human hepatocytes (Author response image 1).

**Author response image 1. sa2fig1:** Seahorse assay showing mitochondrial oxygen consumption rate (OCR) in primary human hepatocytes exposed to APAP (24 h) as compared to baseline (BL) control cells (N=5/group). IgG, isotype antibody control; X209, anti-Il11RA antibody.

Reviewer Recommendations for the authors:The paper will have a high impact on the field, as it identifies a new mechanism for the effects of IL6 on liver regeneration. The techniques and methodology are also sound and multiple approaches are utilized to answer the main question of the paper.1) The major limitation of this study is that the broad conclusion about "overturning the paradigm that Il6 promotes liver regeneration" is not convincingly supported by the presented data. The manuscript is mostly focused on the injury phase of APAP hepatotoxicity (6 and 24 hr post-APAP) and provides limited information on the peak regeneration phase (i.e. later time points – e.g. 48 and 72 hr post-APAP).

We agree with the reviewer that the original title was overstated and we have amended it, and the manuscript throughout, to limit our claims. The new title reads: “The pro-regenerative effects of HyperIL6 in drug induced liver injury are unexpectedly due to competitive inhibition of IL11 signaling”.

While regeneration is active at 24 hours we agree that it is more pronounced at 48 hours and molecular markers of regeneration will be even more apparent at later time points. We have now performed additional experiments that provide further confirmatory data at the 48 h time point (revised Figure 3). Indeed, markers of regeneration are more pronounced at the 48 h time point and complement the 24 h data. This is particularly noticeable for Ki67 (new data) where limited Ki67 staining is apparent in X209-treated or HyperIl6-expressing livers at 24 h (Figure 2—figure supplement 1), whereas robust staining (again STAT3-independent) is seen at 48 h (revised Figure 3K). Cyclin D1 expression is also more apparent at the 48 h time point (revised Figure 3I). In these new experiments we also included additional DMSO controls, to rule out any putative confounding vehicle effects.

Further, this is complicated by the fact that liver regeneration in APAP model is a compensatory phenomenon secondary to liver injury. Thus, any effect on liver injury (as observed in this manuscript) will indirectly alter liver regeneration response. Thus, it is hard to interpret about direct role of Il6 in liver regeneration in the current study. Overall, it is incorrect to interpret in general about role of Il6 in regeneration based on the limited data in the APAP model presented in this study.

We agree that it was an overreach in our interpretation of the data to discuss the findings so generally. In general the degree of liver regeneration is proportional to the degree of liver injury until such a time as regeneration cannot restore liver mass / repair the liver damage. Our manuscript addresses the published and accepted role of HyperIl6 on liver regeneration from which multiple investigators have made the extension to IL6 itself. As an example see the manuscript entitled “*The regenerative activity of interleukin-6*” (Galun and Rose-John 2013). We have been careful in the revision not to overstate our findings and also to focus on effects of HyperIL6, not IL6 (as per the revised title), but do discuss our findings in the context of the published literature.

Cleaner model such as partial hepatectomy can be utilized for such interpretations.

This manuscript concerns HyperIL6 effects in APAP-induced liver injury. The study of drug-induced liver injury is made clear in the revised title and text and we thank the Reviewer for helping us sharpen our focus. After our initial discovery of the unexpected effects of HyperIL6 on IL11 signaling we think it possible that additional studies of the liver (e.g. in partial hepatectomy or CCl4 toxicity) and of other organs where HyperIL6 has a regenerative effect (e.g. kidney, optic nerve or spinal cord), may be performed. However, this does not impact the current manuscript.

Further, it will be incorrect to interpret physiological role of Il6 solely based on experiments utilizing HyperIL6 overexpression.

We agree and make this point in the manuscript. In the revision we have made it clear that the study is of HyperIL6. Indeed, IL6 administration to mice with liver injury does not promote liver regeneration whereas HyperIL6 does (Nechemia-Arbely et al., 2011), which we now discuss (see revised conclusion).

In this regard authors should discuss important previous study by James et al., utilizing Il6 KO mice in APAP model which showed that despite no effect on initial APAP-induced liver injury, Il6 KO mice displayed reduced liver regeneration and slower recovery (PMID: 13679052).

We are glad to include this manuscript (James et al., 2003) in our revision. We had already cited the seminal paper from Science showing that IL6 KO mice had impaired regeneration (Cressman et al., 1996) and also a contradictory manuscript (Sakamoto et al., 1999), which showed otherwise. The revised manuscript has a fuller discussion of these studies.

2) In Figure 3, in order to demonstrate that protective effects of HyperIL6 are independent of STAT3, authors used STAT3 inhibitor (iSTAT3). Authors investigated only single time point (6 hour post-APAP), which does not provide information if stimulation of liver regeneration by hyperIL6 is independent of STAT3. In order to study liver regeneration, authors should study later time points such as 48 hr post-APAP or at least 24 hr time point as done for other figures in the paper.

In revised manuscript we studied the effects of iSTAT3 in APAP-damaged hepatocytes and livers across a range of timepoints. in vitro, we show effects at 24 h post APAP (revised Figure 1G, I and J) and in vivo, we show effects at 6 h (revised Figure 3A-E), and at 48 h (revised Figure 3F-N). We believe this, in particular the new 48 h time point in vivo, addresses the Reviewer’s comment.

Further, it is not clear from the manuscript if vehicle control group for iSTAT3 was utilized in this experiment. From in-vitro experiments presented earlier in the manuscript, I assume DMSO was the vehicle for iSTAT3. If that's the case, proper vehicle control is required as DMSO is known to inhibit Cyp2e1 and APAP metabolism to toxic metabolite and thus show protective effects against APAP toxicity.

The reviewer is correct that DMSO was used as control in the in vitro experiments and we saw no effect, as shown in the original manuscript. In our in vivo experiments, S31-201 was dissolved in DMSO as a stock solution and this was further diluted in saline (1:10) prior to injection 16 h prior to APAP. In earlier studies, which we now cite (Park et al., 1988), DMSO injected at a dose of 1ml/kg, 4 h prior to APAP was shown to reduce liver damage.

To address the Reviewer’s point, we have performed new experiments comparing DMSO only to a variety of experimental conditions (baseline, HyperIl6 expression ± iSTAT, anti-IL11RA) at a 48 h time point (revised Figure 3). This shows that DMSO has no effect on APAP-induced liver damage at any level of assessment in our model. This might reflect the lower amount of DMSO we administer (0.2ml/kg as compared to 1ml/kg used by Park et al.,), the time of administration (16 h prior to APAP, as compared to 4 h), the amount of APAP given (400mg/kg, as compared to 250mg/kg) or strain effects (we use C57BL/6NTac, as compared to BALB/c) (Widjaja et al., 2021).

For completeness, in Author response image 2, we also examined the effects of DMSO on APAP-induced liver damage at 24 h in our model. In keeping with the data presented in the manuscript (48 h), we observed no effect of DMSO at 24 h.

**Author response image 2. sa2fig2:** Serum ALT and AST levels in APAP-injected mice treated with or without DMSO. DMSO (0.2ml/kg) was administered 16 h prior to APAP injection; control mice received saline injection; mice were harvested 24 h post injection. N=5 mice/group. Data are shown as box-and-whisker with median (middle line), 25th–75th percentiles (box) and min-max values (whiskers), one-way ANOVA with Tukey’s correction.

3) This study does provide interesting data on the role of IL11 in APAP-induced liver injury but with limited mechanistic details. Based on these findings, more comprehensive and mechanistic studies in future will be helpful to delineate how IL11 regulate APAP-induced liver injury. It will also be important to rule out any impact of Il11 deletion on metabolic activation of APAP.

While we have been working on this revision, we published a separate study on the mechanisms underlying IL11 effect in the APAP-induced liver injury that are centered on ERK, NOX4 and *JNK* that we now reference and discuss (Widjaja et al., 2021). For the Reviewer’s information, loss-of-function IL11 or IL11RA has no effect on APAP metabolism and instead prevents cellular injury downstream of NAPQI accumulation. We refer the Reviewer to the published manuscript for more detail.

References

Cressman, D. E., L. E. Greenbaum, R. A. DeAngelis, G. Ciliberto, E. E. Furth, V. Poli, and R. Taub. 1996. “Liver Failure and Defective Hepatocyte Regeneration in Interleukin-6-Deficient Mice.” Science 274 (5291): 1379–83.

Galun, Eithan, and Stefan Rose-John. 2013. “The Regenerative Activity of Interleukin-6.” Methods in Molecular Biology 982: 59–77.

James, Laura P., Laura W. Lamps, Sandra McCullough, and Jack A. Hinson. 2003. “Interleukin 6 and Hepatocyte Regeneration in Acetaminophen Toxicity in the Mouse.” Biochemical and Biophysical Research Communications 309 (4): 857–63.

Nechemia-Arbely, Yael, Anat Shriki, Ulrich Denz, Claudia Drucker, Jürgen Scheller, Jonathan Raub, Orit Pappo, Stefan Rose-John, Eithan Galun, and Jonathan H. Axelrod. 2011. “Early Hepatocyte DNA Synthetic Response Posthepatectomy Is Modulated by IL-6 Trans-Signaling and PI3K/AKT Activation.” Journal of Hepatology 54 (5): 922–29.

Ng, Benjamin, Anissa A. Widjaja, Sivakumar Viswanathan, Jinrui Dong, Sonia P. Chothani, Stella Lim, Shamini G. Shekeran, et al. 2021. “Similarities and Differences between IL11 and IL11RA1 Knockout Mice for Lung Fibro-Inflammation, Fertility and Craniosynostosis.” Scientific Reports 11 (1): 14088.

Park, Y., R. D. Smith, A. B. Combs, and J. P. Kehrer. 1988. “Prevention of Acetaminophen-Induced Hepatotoxicity by Dimethyl Sulfoxide.” Toxicology 52 (1-2): 165–75.

Sakamoto, T., Z. Liu, N. Murase, T. Ezure, S. Yokomuro, V. Poli, and A. J. Demetris. 1999. “Mitosis and Apoptosis in the Liver of Interleukin-6-Deficient Mice after Partial Hepatectomy.” Hepatology 29 (2): 403–11.

Widjaja, Anissa A., Jinrui Dong, Eleonora Adami, Sivakumar Viswanathan, Benjamin Ng, Leroy S. Pakkiri, Sonia P. Chothani, et al. 2021. “Redefining IL11 as a Regeneration-Limiting Hepatotoxin and Therapeutic Target in Acetaminophen-Induced Liver Injury.” Science Translational Medicine 13 (597). https://doi.org/10.1126/scitranslmed.aba8146.